# COVID-19 Vaccination Behavior of People Living with HIV: The Mediating Role of Perceived Risk and Vaccination Intention

**DOI:** 10.3390/vaccines9111288

**Published:** 2021-11-06

**Authors:** Li Qi, Li Yang, Jie Ge, Lan Yu, Xiaomei Li

**Affiliations:** 1School of Nursing, Health Science Centre, Xi’an Jiaotong University, Xi’an 710049, China; liqi@qmu.edu.cn; 2School of Nursing, Qiqihar Medical University, Qiqihar 161006, China; 3School of Nursing, Qingdao University, Qingdao 266073, China; yangli81@qdu.edu.cn; 4School of Public Health, Qiqihar Medical University, Qiqihar 161006, China; gejie@qmu.edu.cn; 5AIDS and STD Institute, Heilongjiang Center for Disease Control and Prevention, Harbin 150030, China; lanyu200406@yahoo.com

**Keywords:** people living with HIV, COVID-19 vaccination behavior, perceived risk, behavioral intention, mediating role

## Abstract

The COVID-19 vaccination behavior of people living with HIV (PLWH) was examined via a cross-sectional web-based survey of PLWH aged 18 years and older. The survey was conducted from l May to 20 June 2021. The survey included social demographic information; vaccination behavior (B); and questions related to perceived usefulness (PU), perceived risk (PR), subjective norms (SNs), perceived behavior control (PBC), and behavior intention (BI). The associations between the questionnaire variables and COVID-19 vaccination behavior were assessed by calculating the descriptive data, correlation analysis, and structural equation modeling. In total, 43.71% of the 350 eligible respondents had received a COVID-19 vaccine. The differences in COVID-19 vaccination behavior according to age, gender, religious belief, marital status, income, education level, and occupation were not obvious (*p* > 0.05). PU had a significantly negative effect on PR (*p* < 0.05). PR had a significantly negative effect on BI (*p* < 0.05). SNs had a significantly positive effect on BI (*p* < 0.05). BI had a significantly positive effect on B (*p* < 0.05). PR fully mediated the effects of PU on BI, BI fully mediated the effects of PR on B, and BI fully mediated the effects of SNs on B (*p* < 0.05). Health policymakers and medical workers should provide more information about the risks of vaccine application to improve the vaccination behavior of PLWH.

## 1. Introduction

The global epidemic of coronavirus disease 2019 (COVID-19) is the most widespread and influential public health event of recent years [1,2,3,4]. The global data available from the WHO website on 9 July 2021 reported 185,038,806 confirmed COVID-19 cases and 4,006,882 deaths [5]. Vaccination is one of the most effective and cost-effective health measures that can be used to prevent COVID-19 [6,7,8]. Efficient COVID-19 vaccination delivery with a high population coverage is the only foreseeable means of generating herd immunity and controlling and preventing COVID-19 [9,10,11,12].

On 29 March, the National Health Commission of the People’s Republic of China issued the “Technical Vaccination Recommendations for COVID-19 Vaccines in China (First Edition)” [13], which cleared the COVID-19 vaccine for administration to people aged 18 and above and provided vaccination suggestions for people aged 60 and above, chronic disease patients, etc. The guidelines recommended that chronic disease patients with stable health and good drug control should be vaccinated voluntarily after weighing the advantages and disadvantages. However, there are no data on the safety and efficacy of the COVID-19 vaccine for people living with human immunodeficiency virus (HIV)/acquired immunodeficiency syndrome (AIDS) (PLWHA). People living with HIV (PLWH) account for approximately 0.5% of the global population [14,15]. The vaccination of PLWH also affects vaccine coverage. Some scholars believe that PLWH should improve their vaccination rate as soon as possible [16]. Because there are not enough data on the effects of and adverse reactions to the vaccine among PLWH, their vaccination attitudes will affect their vaccination behavior.

The aim of this study was to establish a theoretical model that explains the mediating effect of vaccination intention regarding the COVID-19 vaccine on the vaccination behavior of PLWH in China by developing a structural equation model that comprehensively demonstrates the correlations among the influencing factors. Our findings provide basic data for developing COVID-19 vaccine education programs and interventions targeting vaccine hesitancy among PLWH.

## 2. Literature Review and Research Hypotheses

Rahimi et al. [17,18] contended that the user’s subjective perceived usefulness (PU) and perceived ease of use (PE) affect their behavioral intention (BI) and behavior (B). The concept of perceived risk (PR) was originated in the field of psychology by Bauer of Harvard University. He believed that the purchase behavior of consumers may not be able to indicate whether the expected results are correct, and some results may make consumers unhappy. Therefore, uncertainty about the results is implicit in consumers’ purchase decisions [19,20]. Ajzen found that people’s behavior is not completely voluntary but is under control. Behavior attitude, subjective norms (SNs), and perceived behavior control (PBC) together affect BI, while BI and PBC affect actual behavior (B) [21,22,23]. The PE for the COVID-19 vaccine mainly depends on the time, place, and price of the vaccination. At present, in order to speed up the process of vaccination, China has set up many vaccination stations providing free vaccinations. In some places, in order to facilitate the vaccination of residents, vaccinations are carried out at the workplace, where residents gather relatively frequently. Therefore, this study did not consider PE. To examine the influences on B, we set up the following research framework (see Figure 1) and research hypotheses.

**Hypothesis** **1** **(H1).***PU exerts a negative effect on PR*.

**Hypothesis** **2** **(H2).***PU exerts a positive effect on BI*.

**Hypothesis** **3** **(H3).***PR exerts a negative effect on BI*.

**Hypothesis** **4** **(H4).***SN exerts a positive effect on BI*.

**Hypothesis** **5** **(H5).***PBC exerts a positive effect on BI*.

**Hypothesis** **6** **(H6).***PBC exerts a positive effect on B*.

**Hypothesis** **7** **(H7).***BI exerts a positive effect on B*.

## 3. Methods

### 3.1. Study Design

We conducted a cross-sectional anonymous web-based survey using an electronic questionnaire, distributed via online social platforms (WeChat and QQ) among PLWH (i.e., PLWH who were 18 years or older and had good drug control). The survey was conducted between l May and 20 June 2021. When PLWH came for antiretroviral treatment (ART) drugs and a physical examination, they scanned the QR code of the questionnaire and then filled in the questionnaire. At the same time, we also used the snowball method to spread and distribute the questionnaire among PLWH.

Before we presented the questionnaire to PLWH, the questionnaire was pilot-tested by a panel of experts in related fields, including an expert in infectious diseases, a behavioral psychologist, an epidemiologist, and a statistician. Specifically, the experts proofread the questionnaire and ascertained its content validity in terms of the fit between each statement in the questionnaire and the corresponding theoretical variable. The questionnaire was then amended according to the suggestions for revision made by the experts.

### 3.2. Questionnaire

The questionnaire consisted of the following sections: (1) sociodemographic predictor variables, which included age, gender, religious belief, marital status, income, education level, and occupation; HIV related characteristics, which included duration of diagnosis, chronic disease, virus load detection, and the side effects of antiretroviral drugs; (2) perceived usefulness (PU), perceived risk (PR), subjective norms (SNs), perceived behavior control (PBC), and vaccination behavioral intention (BI) regarding the COVID-19 vaccine (see Table 1 for details); (3) and COVID-19 vaccination behavior (B). The questionnaire included 31 questions and generally took less than 10 min to complete.

### 3.3. Variables and Measurements

PU, PR, SNs, PBC, and BI were scored on a 5-point Likert scale, with a score of 1 indicating strong disagreement and a score of 5 indicating strong agreement. B was transformed to a binary variable (1 = yes and 0 = no) in order to facilitate analysis. 

### 3.4. Reliability of the Questionnaire

Cronbach’s alpha internal reliability method produced a figure of 0.858 for the internal consistency of our research, which showed that this study instrument was valid and reliable for data-gathering activities.

### 3.5. Statistical Analysis

The software packages SPSS 23.0 and Analysis of Moment Structures (AMOS) 20.0 were used for statistical analysis. Calculation of the descriptive data, correlation analysis, and structural equation modeling (SEM) were conducted. A *p*-value of <0.05 was considered statistically significant.

### 3.6. Ethical Considerations

This study was approved by the Ethics and Research Review Committee of Qiqihar Medical University ([2020]38).

## 4. Results

### 4.1. Intention to Get Vaccinated, Vaccination Status, and Participant Characteristics

Overall, 350 respondents completed the survey, 95.7% of whom were male (*n* = 335). The age (mean ± SD) of the respondents was 36.01 ± 9.64 years. Of these, 88.3% had no religious beliefs (*n* = 309); 76.6% were single (*n* = 268); 70.9% had an income below RMB 5000 (*n* = 248); and 57.7% had a college degree, a bachelor’s degree, or above (*n* = 202). In total, 6.0% were medical-related majors (*n* = 21), while 55.1% were service trade staff (*n* = 193). The differences according to age, gender, religious belief, marital status, income, education level, and occupation were not statistically significant (*p* > 0.05). Among the respondents, 64.0% *(n* = 224) had been diagnosed with an HIV infection for less than five years, and 19.4% (*n* = 68) had been diagnosed with other chronic diseases. In total, 189 respondents knew their viral load results; of these, 95.2% (*n* = 180) did not detect the viral load. After taking antiretroviral drugs, 6.0% (*n* = 21) of the respondents had no side effects, 80.6% (*n* = 282) had mild side effects, and 13.4% (*n* = 47) had moderate side effects. The differences in the duration of diagnosis, the presence of other chronic diseases, the virus load detection, and the side effects of antiretroviral drugs were not statistically significant (*p* > 0.05) (see Table 2 and Table 3 for details).

### 4.2. The SEM Fitting Index Results

The chi-square/degrees of freedom (*χ*^2^/DF) and root mean square error of approximation (RMSEA) were used to test the fitness of the SEM, where 1 < *χ*^2^/DF < 3 and RMSEA < 0.05 indicate a better fit [24,25]. The Tucker–Lewis index (TLI) and the comparative fit index (CFI) were the incremental fit indices, where TLI > 0.95 and CFI > 0.95 indicate a very good fit [26]. In our study, the *χ*^2^/DF was 1.318, the RMSEA was 0.031, the TLI was 0.961, and the CFI was 0.964. Therefore, the overall fit of this research model was acceptable.

### 4.3. Model Analysis Results

The path analysis results revealed that PU had a significantly negative effect on PR, thus supporting H1. PR had a significantly negative effect on BI, thus supporting H3. SNs had a significantly positive effect on BI, thus supporting H4. BI had a significantly positive effect on B, thus supporting H6. The hypothesis test results are presented in Table 4. The results of deleting meaningless paths are shown in Figure 2.

As shown in Table 5, PR fully mediated the effects of PU on BI, BI fully mediated the effects of PR on B, and BI fully mediated the effects of SNs on B.

## 5. Discussion

The results showed that 153 respondents to the questionnaire (43.71%) had been vaccinated against COVID-19. Previous studies on vaccination against COVID-19 in PLWH mainly reflected vaccination willingness and COVID-19 vaccine hesitancy. In one study, 28.7% of the participants declared their hesitancy about being vaccinated against COVID-19 [16]. More studies on vaccination intention have focused on the general population. A survey on the attitudes toward COVID-19 vaccination among the general population showed that 82.94% of the participants wished to be vaccinated, 14.73% of the participants were hesitant, and only 2.33% of the participants refused to be vaccinated [27]. A comparison between the data from these two studies suggested that there was a more serious phenomenon of vaccine hesitancy among PLWH. However, because these data came from different populations, it was impossible to conclude that the vaccine hesitation of PLWH was more serious than that of other groups. At present, COVID-19 vaccination is still in progress, the vaccination situation is still changing, and people’s vaccination intentions and behavior will also continue to change.

We found that there were differences regarding vaccination intention and the COVID-19 vaccination status of the respondents according to age, gender, religious belief, marital status, income, education level, occupation, the duration of HIV diagnosis, the presence of other chronic diseases, the virus load detection, and the side effects of antiretroviral drugs, but these differences were not obvious (*p* > 0.05). However, until the end of the survey period, most research on COVID-19 vaccination had focused on vaccination attitudes. One Chinese study showed that vaccine refusal and vaccine hesitancy were not affected by age, marital status, income, or occupation but were dependent on gender and education level among residents in Guangzhou [8]. However, another recent Chinese study showed that vaccine hesitancy and vaccination refusal were associated with gender [27]. They found that women were more likely to be hesitant about vaccines [27,28,29]. The reasons for these gender differences are unclear. On the one hand, women generally pay more attention to their health, and their awareness of healthcare is higher than that of men [30]. On the other hand, the results might have been affected by regional differences and cultural differences [31,32,33]. In our study, there was no significant gender difference in vaccination behavior. This might be a result of the small number of female respondents included in our study. The vaccine behavior differences according to the duration of HIV diagnosis were not obvious (*p* > 0.05). This might be related to the effective drug control of PLWH. In our study, the intention to get vaccinated and the vaccination behavior toward COVID-19 were not shown to be associated with virus load. This might be related to the fact that only 54.0% of our participants knew their viral load, and fewer had had their viral load detected. The difference in vaccination intention among the general adult population between those who had been diagnosed with combined chronic diseases and those who had not was statistically significant [34]. However, we found that for PLWH, the difference in vaccination intention between those who had been diagnosed with other chronic diseases and those who had not was not statistically significant. This might be related to their HIV status, since they might be more concerned about HIV control. We also found that the differences in vaccination intention related to the side effects of taking antiretroviral drugs were not obvious (*p* > 0.05). Clinically, mild side effects can be eliminated without special treatment, moderate side effects can be controlled by symptomatic treatment, and clinicians will change the treatment scheme to reduce the impact of drug side effects when the drug side effects of PLWH are very serious. The lack of any obvious differences in vaccination intention here might be related to the controllable nature of the side effects of the drugs used by our participants.

Our results showed that PU had a negative effect on PR, PR had a negative effect on BI, SNs had a positive effect on BI, and BI had a positive effect on B. The PR of PLWH regarding COVID-19 vaccination played a mediating role between PU and BI. This means that PU did not directly affect BI but affected BI through PR. Our results also showed that BI played a mediating role between PR and vaccination behavior. That is, PR did not directly affect vaccination behavior, but affected vaccination behavior through BI. These results are similar to those obtained in other studies, which have shown that risk perception is a critical determinant of behavioral intention and health behavior [35,36]. The time from research to use of the COVID-19 vaccine was short, and the vaccine has been approved for emergency use. However, long-term data on the vaccine’s efficacy and safety still need to be continuously monitored [37]. Many people are still worried about the safety and side effects of the vaccine [38]. In an uncertain situation, people’s behavioral choices tend to reduce the perceived risk rather than maximizing the perceived benefits [39]. Therefore, in the case of determining the safety of the COVID-19 vaccine, people will consider the effectiveness of the vaccine [40]. Therefore, information regarding the safety and side effects of vaccination should be presented in a timely fashion, and behavioral intention can be increased by reducing the perceived risk of the COVID-19 vaccination. Studies have shown that providing negative information about the COVID-19 pandemic can help to enhance the public’s risk perception, which provides an opportunity to improve the overall COVID-19 vaccination rate [41].

We also found that BI plays a mediating role between SNs and vaccination behavior, but PBC does not affect BI or vaccination behavior. This means that SNs can influence the vaccination rate through behavioral intentions, but PBC does not affect BI, nor does PBC affect vaccination behavior. The result of the mediating role of BI further confirms research on the theory of planned behavior (TPB), which holds that human behavior is not 100% voluntary, but is instead under control [20]. However, according to the TPB, PBC influences behavior directly and indirectly through BI. This conclusion has not been confirmed in our study. This may be related to the lack of data concerning COVID-19 vaccination among PLWH. During the clinical trials and administration of the COVID-19 vaccine, there was limited a amount of data on the usefulness and side effects of the vaccine among PLWH. In the absence of accurate data showing that COVID-19 vaccines are useful to PLWH, they are more concerned about the risk of vaccination. This affects their vaccination behavior through their behavioral intention. Therefore, more data on the usefulness and risks of the COVID-19 vaccine for PLWH are needed to improve the vaccination intention of PLWH and promote the vaccine.

## 6. Limitations

There were several limitations to this survey. Convenient sampling and snowballing were used in the survey, but random sampling was not conducted, which may have affected the representativeness of the research samples. For example, the ratio of male to female PLWH in the survey area was 11.9:1, with fewer women in the survey. This is the main defect of this study and one of the most important problems to be solved in our follow-up investigation. With the changes in COVID-19 and the continued promotion of the COVID-19 vaccination, the awareness of and vaccination intention regarding the COVID-19 vaccine are also changing constantly. Therefore, the vaccination behavior of PLWH needs to be investigated at different stages of the COVID-19 pandemic.

## 7. Conclusions

Our evidence suggests that the behavioral intentions regarding COVID-19 vaccination among PLWH play a mediating role between the perceived risk, subjective norms, and vaccination behavior. The perceived risk plays a mediating role between the perceived usefulness and the behavioral intention. Therefore, scientific popularization should be strengthened to enhance the awareness and the perceived usefulness of the COVID-19 vaccine and reduce the perceived risk of the vaccine for PLWH, thus improving vaccination intention and vaccination behavior in order to achieve vaccine protection.

## Figures and Tables

**Figure 1 vaccines-09-01288-f001:**
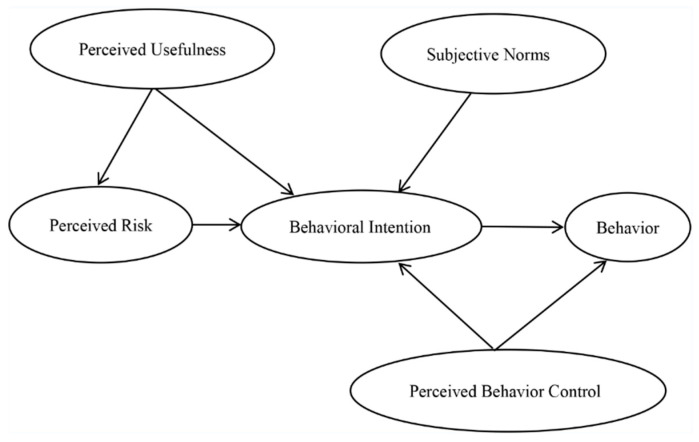
The research framework.

**Figure 2 vaccines-09-01288-f002:**
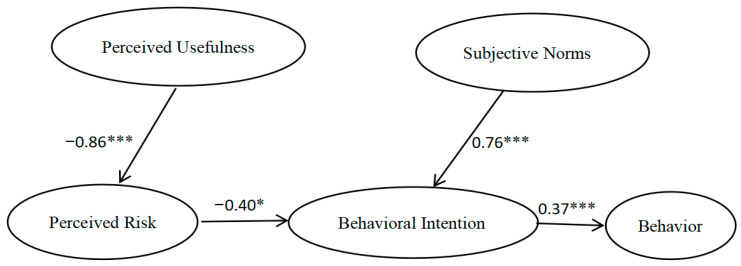
Results of deleting meaningless paths. Note: *** *p* < 0.001; * *p* < 0.05.

**Table 1 vaccines-09-01288-t001:** PU, PR, SN, PBC, and BI.

Research Constructs	Measurement Items
PU	1. You think the COVID-19 vaccine can prevent COVID-19. 2. You think it’s easier to get COVID-19 without vaccination. 3. You think vaccination can benefit you. 4. You think vaccination can benefit others.
PR	1. You think the COVID-19 vaccine is safe. 2. You think the COVID-19 vaccine will have side effects. 3. You think you can be infected with COVID-19, even if you have been vaccinated. 4. You think not vaccinating will have an impact on the people around you.
SN	1. The people around you have been vaccinated. 2. Your family supports your vaccination. 3. You accept your family’s advice regarding the COVID-19 vaccination. 4. You accept your friends’ advice regarding the COVID-19 vaccination. 5. You accept medical workers’ advice regarding the COVID-19 vaccination. 6. You accept the government’s advice regarding the COVID-19 vaccination. 7. You accept the suggestions of media publicity regarding the COVID-19 vaccination. 8. You will get vaccinated after a lot of people have been vaccinated.
PBC	1. You can decide for yourself to get vaccinated. 2. You can choose your own type of vaccine. 3. You can choose your own time for the vaccine. 4. You can choose your own place for the vaccine.
BI	1. You would like to be vaccinated. 2. You support the application of vaccines in PLWH. 3. You will recommend vaccinations to others.

**Table 2 vaccines-09-01288-t002:** Characteristics of participants by intention to get vaccinated and vaccination behavior against COVID-19 (*n* = 350).

	All Subjects (*n* = 350) N(%)	Intention to Get Vaccinated Against COVID-19			COVID-19 Vaccination Behavior		
		Intend to Get Vaccinated (*n* = 280) N(%)	Do Not Intend to Get Vaccinated (*n* = 70) N(%)	*p*-Value	Vaccinated (*n* = 153) N(%)	Do Not Vaccinate (*n* = 197) N(%)	*p*-Value
Sociodemographic							
Gender				0.741			0.174
Male	335(95.7)	269(80.3)	66(19.7)		149(44.5)	186(55.5)	
Female	15(4.3)	11(73.3)	4(26.7)		4(26.7)	11(73.3)	
Age group				0.648			0.198
18–20	4(1.1)	4(100.0)	0(0.0)		3(75.0)	1(25.0)	
21–30	110(31.5)	88(80.0)	22(20.0)		48(43.6)	62(56.4)	
31–40	141(40.3)	108(76.6)	33(23.4)		60(42.6)	81(57.4)	
41–50	64(18.3)	55(85.9)	9(14.1)		33(51.6)	31(48.4)	
51–60	27(7.7)	22(81.5)	5(18.5)		9(33.3)	18(66.7)	
61+	4(1.1)	3(75.0)	1(25.0)		0(0.0)	4(100.0)	
Religious belief				0.454			0.718
Religious belief	41(11.7)	31(75.6)	10(24.4)		19(46.3)	22(53.7)	
No religious belief	309(88.3)	249(80.6)	60(19.4)		134(43.4)	175(56.6)	
Marital status				0.900			0.328
Single	268(76.6)	214(79.9)	54(20.1)		121(45.1)	147(54.9)	
Married	82(23.4)	66(80.5)	16(19.5)		32(39.0)	50(61.0)	
Income				0.852			0.610
≤3000	141(40.3)	111(78.7)	30(21.3)		57(40.4)	84(59.6)	
3001–5000	107(30.6)	85(79.4)	22(20.6)		46(43.0)	61(57.0)	
5001–10,000	74(21.1)	60(81.1)	14(18.9)		36(48.6)	38(51.4)	
>10,000	28(8.0)	24(85.7)	4(14.3)		14(50.0)	14(50.0)	
Educational level				0.439			0.944
Junior high school and below	87(24.9)	70(80.5)	17(19.5)		36(41.4)	51(58.6)	
High school or polytechnic school	61(17.4)	46(75.4)	15(24.6)		26(42.6)	35(57.4)	
College or bachelor degree	186(53.1)	149(80.1)	37(19.9)		84(45.2)	102(54.8)	
Master degree or above	16(4.6)	15(93.8)	1(6.3)		7(43.8)	9(56.3)	
Occupation				0.130			0.742
Medical-related majors	21(6.0)	15(71.4)	6(28.6)		10(47.6)	11(52.4)	
Staff of relevant government departments or teacher	50(14.3)	45(90.0)	5(10.0)		26(52.0)	24(48.0)	
Worker	55(15.7)	45(81.8)	10(18.2)		24(43.6)	31(56.4)	
Farmer	31(8.9)	21(67.7)	10(32.3)		13(41.9)	18(58.1)	
Service trades staff	193(55.1)	154(79.8)	39(20.2)		80(41.5)	113(58.5)	
HIV related characteristics							
Duration of diagnosis				0.126			0.857
≤5 years	224(64.0)	185(82.6)	39(17.4)		100(44.6)	124(55.4)	
6–10 years	98(28.0)	74(75.5)	24(24.5)		41(41.8)	57(58.2)	
11–15 years	19(5.4)	13(68.4)	6(31.6)		9(47.4)	10(52.6)	
16–20 years	7(2.0)	7(100.0)	0(0.0)		3(42.9)	4(57.1)	
>20 years	2(0.6)	1(50.0)	1(50.0)		0(0.0)	2(100.0)	
Chronic Disease				0.418			0.119
Chronic disease	68(19.4)	52(76.5)	16(23.5)		24(35.3)	44(64.7)	
No chronic disease	282(80.6)	228(80.9)	54(19.1)		129(45.7)	153(54.3)	
The side effect of anti-retroviral drugs				0.201			0.091
No side effects	21(6.0)	19(90.5)	2(9.5)		14(66.7)	7(33.3)	
Mild side effects	282(80.6)	227(80.5)	55(19.5)		119(42.2)	163(57.8)	
Moderate side effects	47(13.4)	34(72.3)	13(27.7)		20(42.6)	27(57.4)	

Note: Mild side effects can be eliminated without special treatment; moderate side effects can be controlled by symptomatic treatment.

**Table 3 vaccines-09-01288-t003:** Viral load by intention to get vaccinated and vaccination behavior against COVID-19 (*n* = 189).

Viral Load	All Subjects (*n* = 189) N(%)	Intention to Get Vaccinated against COVID-19			COVID-19 Vaccination Behavior		
		Intend to Get Vaccinated (*n* = 136) N(%)	Do Not Intend to Get Vaccinated (*n* = 53) N(%)	*p*-Value	Vaccinated (*n* = 77) N(%)	Do Not Vaccinate (*n* = 112) N(%)	*p*-Value
Not detected	180(95.2)	131(72.8)	49(27.2)	0.458	74(41.1)	106(58.9)	0.908
detected	9(4.8)	5(55.6)	4(44.4)		3(33.3)	6(66.7)	

Note: A total of 161 of the respondents living with HIV did not detect or did not know their viral load.

**Table 4 vaccines-09-01288-t004:** Hypothesis test results.

Hypothesis	Path between	Nonstandard Coefficient	Standardization Coefficient	S.E.	C.R.	*p*
H1	PU→PR	−1.049	−0.857	0.147	−7.137	***
H2	PU→BI	0.074	0.055	0.306	0.242	0.809
H3	PR→BI	−0.448	−0.404	0.085	−2.025	0.043 *
H4	SN→BI	0.731	0.760	0.055	13.378	***
H5	PBC→BI	0.063	0.090	0.033	1.878	0.060
H6	PBC→B	0.004	0.010	0.021	0.191	0.848
H7	BI→B	0.224	0.370	0.032	7.018	***

Note: *** *p* < 0.001; * *p* < 0.05.

**Table 5 vaccines-09-01288-t005:** Mediating effect test.

Mediation Path			Mediating Effect								Mediating Effect
IV	M	DV	Effect Value	SE	Bias-Corrected 95% CI			Percentile 95% CI			
					Lower	Upper	*p*	Lower	Upper	*p*	
PU	PR	BI	0.405	0.099	0.225	0.612	0.001	0.226	0.613	0.001	Full
PR	BI	B	−0.177	0.044	−0.271	−0.100	0.001	−0.264	−0.095	0.001	Full
SN	BI	B	0.287	0.061	0.161	0.396	0.001	0.155	0.394	0.001	Full

Note: IV = independent variable; M = mediator; DV = dependent variable.

## Data Availability

The datasets generated during the current study are not publicly available but are available from the corresponding author on reasonable request.

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
