# Peer review of "COVID-19 Vaccination Behavior of People Living with HIV: The Mediating Role of Perceived Risk and Vaccination Intention"

_vaccines, 2021, doi:10.3390/vaccines9111288_

Round 1

Reviewer 1 Report

The presented paper is interesting, and in my opinion well written and organized. I have only minor concerns about the work. The points are reported below.

Authors considered data from 350 respondents in high measure male. This set of data should be affected by gender-behaviour? 350 are sufficient for reaching a statistically relevant conclusion. I believe that the sample in large part belonging to one gender it is not representative. Authors should discuss this aspect.

Cronbach’s alpha should be briefly explained in the methods

PLWH should be defined in the abstract

Reviewer 2 Report

The authors present original and exceptionally interesting results on the attitudes of people living with HIV towards COVID-19 vaccination. The study uses a comprehensive approach to the issue and the results are clearly presented. The limitations of the study are presented in an objective manner, particularly the issue of changes in attitudes towards vaccination during different waves of infection. I recommend the study for publication in the present form.  

Author Response

Many thanks for your kind support with our study and the manuscript.

Reviewer 3 Report

This report assesses a small cohort of PLWH for willingness to vaccinate against COVID-19. The proposed relationships between different decision parameters may be helpful in future efforts aimed at improving the vaccination coverage. However, several limitations reduce the impact of the study and need to be addressed/acknowledged.

  1. Pervious studies of uninfected cohorts found gender as one of the major factors that determine the vaccination rate. The authors state that gender was not a factor in their cohort. But over 90% of their subjects were male! It is unclear how they could come to this conclusion on such a small number females.
  2. The focus of this study are PLWH.  However, there is no discussion of how and why HIV infection may affect COVID-19 vaccination decision.
  3. They had the numbers for actual vaccination, but they do not fully utilize this information. Did intention to be vaccinated correlate with actual vaccination? If not, why? Addressing this and other related questions would increase significance of the study.
  4. They use abbreviations (PU, PI, etc) that are not commonly used, so reader needs to constantly refer to the abbreviation list. A more straightforward explanation of the results, especially in the Discussion, would be helpful.

Round 2

Reviewer 3 Report

The authors addressed most of my previous concerns, but one, the effect of HIV infection on decision making, still remains. As I commented before, the manuscript has PLWH in the title, but there is no HIV-related indicators in the analysis. The authors should have clinical information that would allow them to answer the following specific questions.

  1. Did duration of HIV infection affect COVID-19 vaccination decision?
  2. Did HIV disease status (co-morbidities, viral load, anti-retroviral drugs side effects) affect vaccination decision?

Author Response

Response to Reviewer 3 Comments

Point 1: Did duration of HIV infection affect COVID-19 vaccination decision?

Response 1: We deeply appreciate the reviewer’s comments here. Most of the
patients do not know when they have been infected, some of PLWH didn’t detect HIV infection in time after dangerous behavior, some of PLWH accidentally found HIV infection during physical examination, or preoperative examination. Therefore, it is difficult to determine the exact duration of infection, we use the patients diagnosis time of HIV infection to evaluate the duration of HIV infection. In addition, the statistical results show that the duration of diagnosis of PLWH has no effect on vaccination intention and vaccination behavior.

Point 2: Did HIV disease status (co-morbidities, viral load, anti-retroviral drugs side effects) affect vaccination decision?

Response 2: Thank you for your tips on PLWH, we fully agree with your point. We
were also very interested in the impact of the combination of other chronic diseases of PLWH on vaccination intention and vaccination behavior in our study. However, the survey results indicated that the presence or absence of chronic diseases had no effect on vaccination intention and vaccination behavior.

The policy in our study region is that PLWH can voluntarily get a free test of the viral load per year , if they taking anti-retroviral drugs. Therefore, only 189 of the participants took the test and knew their viral load. Moreover, our study confirmed that there was no significant difference in vaccination intention and vaccination behavior between the undetected viral load groups and detected viral load groups.

Most of PLWH have drug side effects after taking anti-retroviral drugs, generally, the side effects of drugs have been controlled or alleviated if treated symptomatically. If the drug side effects of PLWH are very serious, clinicians changed the treatment scheme for PLWH to reduce the impact of drug side effects. Therefore, no PLWH has serious drug side effects in our study, even if they have symptoms of side effects, which are relatively mild and has been controlled after symptomatic treatment. The results showed that the difference of side effects on vaccination intention and vaccination behavior was not statistically significant.

Round 3

Reviewer 3 Report

Thank you for addressing my questions. I suggest that you include all the negative results you mentioned in the manuscript (in the Results and Discussion). One would expect PLWH to be willing to get COVID-19 vaccine, given their encounter with medical treatments and general knowledge of infections. That this is not the case is surprising and needs to be discussed.

Author Response

Thank you very much for your suggestion. We have added this in the Results and Discussion. Before the investigation, we also considered that duration of diagnosis, chronic disease, virus load detection and the side effect of anti-retroviral drugs might affect the vaccination intention and vaccination behavior. But the investigation results showed that the difference of these factors were not statistically significant. Furthermore, this is also one of the problems we want to solve in the follow-up research.